# Stable Gastric Pentadecapeptide BPC 157 as a Therapy for the Disable Myotendinous Junctions in Rats

**DOI:** 10.3390/biomedicines9111547

**Published:** 2021-10-27

**Authors:** Mladen Japjec, Katarina Horvat Pavlov, Andreja Petrovic, Mario Staresinic, Bozidar Sebecic, Matko Buljan, Hrvoje Vranes, Ana Giljanovic, Domagoj Drmic, Miroslav Japjec, Andreja Prtoric, Eva Lovric, Lovorka Batelja Vuletic, Ivan Dobric, Alenka Boban Blagaic, Anita Skrtic, Sven Seiwerth, Sikiric Predrag

**Affiliations:** 1Department of Surgery, School of Medicine, University of Zagreb, 10000 Zagreb, Croatia; jap.mladen@gmail.com (M.J.); ravnateljstvo@kb-merkur.hr (M.S.); bsebecic2@gmail.com (B.S.); andreja2007@gmail.com (A.P.); ivandobricmd@gmail.com (I.D.); 2Department of Pathology, School of Medicine, University of Zagreb, P.O. Box 910, Salata 10, 10000 Zagreb, Croatia; katarina.horvat@gmail.com (K.H.P.); petrovic.andrea@gmail.com (A.P.); eva.lovric@kb-merkur.hr (E.L.); lbatelja@mef.hr (L.B.V.); sven.seiwerth@mef.hr (S.S.); 3Department of Pharmacology, School of Medicine, University of Zagreb, P.O. Box, 916, Salata 11, 10000 Zagreb, Croatia; mbuljan@kbd.hr (M.B.); hrvoje.vranes@gmail.com (H.V.); giljanovic.ana.hr@gmail.com (A.G.); iddrmic@mef.hr (D.D.); miroslav.japjec@gmail.com (M.J.); abblagaic@mef.hr (A.B.B.)

**Keywords:** nitric oxide, oxidative stress, BPC 157 therapy, myotendinous junctions, leg contracture, walking recovery index

## Abstract

(1) Aim: The stable gastric pentadecapeptide BPC 157 is known to heal transected muscle, tendon, and ligament. Thereby, in this study, we investigated the effect of BPC 157 on the dissection of the quadriceps tendon from the quadriceps muscle in rats. (2) Materials and Methods: Myotendinous junction defect, which cannot heal spontaneously in rats, as evidenced with consistent macro/microscopic, biomechanical, functional assessments, *eNOS*, and *COX-2* mRNA levels and oxidative stress and NO-levels in the myotendinous junctions. BPC 157 (10 µg/kg, 10 ng/kg) regimen was given (i) intraperitoneally, first application immediately after surgery, last 24 h before sacrifice; (ii) per-orally, in drinking water (0.16 µg/mL, 0.16 ng/mL, 12 mL/rat/day), till the sacrifice at 7, 14, 28 and 42 postoperative days. (3) Results: These BPC 157 regimens document prominent therapy effects (macro/microscopic, biomechanical, functional much like *eNOS* and *COX-2* mRNA levels and counteracted oxidative stress and NO-levels in the myotendinous junctions), while controls have a poor presentation. Especially, in rats with the disabled myotendinous junction, along with full functional recovery, BPC 157 counteracts muscle atrophy that is regularly progressive and brings muscle presentation close to normal. Accordingly, unlike the perilous course in controls, those rats, when receiving BPC 157 therapy, exhibit a smaller defect, and finally defects completely disappear. Microscopically, there are no more inflammatory infiltrate, well-oriented recovered tissue of musculotendon junction appears in BPC 157 treated rats at the 28 days and 42 days. (4) Conclusions: BPC 157 restores myotendinous junction in accordance with the healing of the transected muscle, tendon, and ligament.

## 1. Introduction

After careful dissecting of quadriceps tendon from the quadriceps muscle, we focused on the myotendinous junction recovery [1,2] and the stable gastric pentadecapeptide BPC 157 therapy [3,4,5,6,7,8,9,10,11,12,13,14,15,16,17] known to heal both transected and detached tendon and transected muscle [18,19,20,21,22], applied alone, as native peptide therapy, that may be effective in rat injury, given intraperitoneally or in drinking water [3,4,5,6,7,8,9,10,11,12,13,14,15,16,17].

Of note, the myotendinous junction, as the weakest element in the muscle-tendon unit, its anatomy, and structural particularities were largely reviewed i.e., [1,2,23]. On the other hand, with the interrelated muscle and tendon healing to recover the muscle-tendon junction injury [1,2] to recover muscle and tendon at the same time unit, the therapy studies upon myotendinous junction injury, are still lacking.

In tendon and muscle healing, the lack of therapy studies upon myotendinous junction injury may be due to the apparent inconsistencies between the suggested various growth factors high physiologic significance [23] and not achieved practical realization of the supposed healing effect [23]. Namely, unlike BPC 157 native peptide therapy [3,4,5,6,7,8,9,10,11,12,13,14,15,16,17], there is a lacking easy practical applicability for many various growth factors that have been suggested to be necessary for natural healing [23]. The illustrative attempts include both highly sophisticated delivery technics (i.e., a suture carrying nanoparticle/pEGFP-basic fibroblast growth factor (bFGF) and pEGFP-vascular endothelial growth factor A (VEGFA) complexes developed to transfer the growth factor genes into injured tendon tissues to promote healing [24], and various growth factors combinations given together [25,26]), and various carriers (i.e., growth factors-loaded collagen sponge; BMP-12 cDNA-transduced muscle grafts addition [25,26]) to demonstrate experimental usefulness for the tendon or muscle repair, but the direct, local delivery of growth factors has limited use [23].

Thereby, for the healed both transected and detached tendon and transected muscle, as native peptide therapy [18,19,20,21,22,27,28], and then the myotendinous junction injury to be healed [1,2], necessitating more advanced strategies for a sustained, safe and reproducible delivery to ascertain the effect, the more general BPC 157 evidence may be also important [3,4,5,6,7,8,9,10,11,12,13,14,15,16,17]. Illustratively, BPC 157 (as an original anti-ulcer peptide, stable in human gastric juice more than 24 h unlike standard growth factors rapidly destroyed, and thereby easy applicable, lethal dose (LD1) not achieved, used to be in ulcerative colitis trials and now multiple sclerosis [3,4,5,6,7,8,9,10,11,12,13,14,15,16,17]), within wound/healing gastrointestinal ulcer relations and angiogenesis is advantageous over standard growth angiogenic factors [7]. That point is especially well-reviewed [7]. BPC 157 is always given alone without any carrier, and within the same dose range, it exhibits ready combining of the wound and ulcer healing in the whole gastrointestinal tract [7]. Contrarily, the standard angiogenic growth factors have a different effect [7]. Their beneficial effect does not involve the whole gastrointestinal tract, and they are regularly applied in wound healing studies with different carriers [7]. Thereby, the myotendinous junction injury healing as a wound healing, different tissues simultaneous healing, requires agents like the stable gastric pentadecapeptide BPC 157, certain effect attribution peptide always given alone without carrier addition [7].

In support, in addition to the healed both transected and detached tendon and transected muscle [18,19,20,21,22], BPC 157 has an especial effect on tendon fibroblasts.

Note, tendon fibroblasts migrate into the injured site, proliferate, and produce different types of collagens and glycoproteins to form the extracellular matrix, avoiding unpleasant ossification [19,22] (i.e., with various factors and different carriers in the bone tunnel are readily produced [29,30]); BPC 157 induced Achilles tendon-to-bone healing: tendon to bone could not be healed spontaneously, but it was recovered by this peptide [19,22]. Thus, with BPC 157, the osteotendinous junction successfully restored may favor the restoration of the other junction [19,22], and likely realize the myotendinous junction restoration. Likewise, BPC 157 healed transected medial collateral ligament in rats [31], and healed pseudoarthrosis in rabbits, and counteracted bone resorption in two distinctive bone lesions models in rats (i.e., femoral head osteonecrosis, periodontitis alveolar bone loss) [32,33,34].

Also, in addition to the transected muscle healing [21], BPC 157 exhibits special muscle healing also after severe crush and denervation [35,36,37] even in conditions of severe healing impairment induced by systemic corticosteroid administration [37]. Finally, BPC 157 fulfills the particular notations considered as important for the myotendinous junction injury to be healed i.e., collagen fragments orientation [23], vascular density at myotendinous junction [38], NO-system activity [15] (if NOS activity is inhibited by competitive inhibitors of NOS, tendon healing is reduced [39]). Commonly, for both transected and detached tendon healing and transected muscle healing [18,19,20,21,22] and other wound healing [7,8,40,41,42,43,44], the stable gastric pentadecapeptide BPC 157 likely controls functions of collagen fragments. In the transected muscle, abundant collagen fibers with longitudinal orientation appear as in the transected tendon healing [19,22]. Likewise, BPC 157 has a particular angiogenic effect along with its healing [18,43,45,46] in transected muscle, an increase of blood vessels well-formed and longitudinally oriented, with desmin staining prominent positivity present throughout the active border of the transection, unlike controls, matching with the effect noted in the transected or detached tendon [18,19,20,21,22]. Also, BPC 157 directly protects endothelium [47,48], alleviates the peripheral vascular occlusion disturbances [49,50,51,52,53,54,55,56,57,58,59,60], rapidly activating alternative bypass pathways. Likewise, BPC 157 possesses strong antioxidant activity [49,50,51,52,53,54,55,56,57,58,59,60], also shown in the tendon tissue, in particular [19,20]. Finally, considering NO-system activity, BPC 157 largely interacts with NO-system [15]. It seems that it modulates its activity [15] since counteracts adverse effects of NOS-blocker and NOS-substrate application [12,49,50,51]. Also, it may induce NO-release by itself [61] (note, when additional NO is added, tendon healing is enhanced [39]).

Lastly, BPC 157 has a wider beneficial effect on muscle function. Namely, BPC 157 counteracts the various muscle disabilities and recovers muscle functioning i.e., after abdominal aorta anastomosis [47], L2–L3 compression [62], severe electrolytes disturbances [63,64,65] stroke [57], application of the succinylcholine and thereby, neuromuscular junction [65], neuroleptics [66,67,68], or neurotoxin (1-methyl-4-phenyl-1,2,3,6-tetrahydrophyridine (MPTP), cuprizone) [46,69]. This also includes the recovery of the various sphincters function [66,70,71,72]. Also, BPC 157 curative effects were ascribed to its interactions with several molecular pathways [4,5,27,28,41,45,56,73,74,75,76], and in particular, BPC 157 counteracts cachexia induced by a tumor in mice [4], muscle wasting, deranged muscle proliferation and myogenesis, an increase of the proinflammatory cytokines i.e., IL-6, TNF-α and the changes in the expression of FoxO3a, p-AKT, p-mTOR, and P-GSK-3β [4].

Therefore, in the rats having careful dissecting of the quadriceps tendon from the quadriceps muscle, we used stable gastric pentadecapeptide BPC 157 once a day intraperitoneal regimen, or per-oral regimen, continuously in drinking water. Here, the positive outcome reestablished tendon-muscle continuity will be supported by the functional, biomechanical, microscopical, and macroscopical findings as well as *eNOS*, *iNOS*, *nNOS*, *COX-2* mRNA levels, and NO- and MDA-levels in the defect formed after dissecting quadriceps tendon from the quadriceps muscle, presenting a healing potential of its own. These may directly affect the healing repair of connective tissues [18,19,20,21,22], presenting NO-system directly related to skeletal muscle and tendon and muscle and tendon injuries, and considering the essential role of NO in healing [39,77] as an ordered multistage process involving inflammation and oxidative stress [78]. There, not particularly evaluated in the mentioned reviews [1,2,23], appears an injured hind limb contracture [19,20,21,22,31,35,36,37,47] rat held upright, as for quantification of functional recovery [79], approaching, at least in principle, the bipedal motion of quiet standing [80].

That rapid counteraction was used as a particular hallmark of the reestablished function and achieved healing recovery in the previous BPC 157 studies [19,20,21,22,31,35,36,37,47].

Finally, considering the native peptide therapy, and described the beneficial effect in muscle and tendon healing [18,19,20,21,22,27,28], it is likely that we can unmistakably attribute all of its effects i.e., functional, biomechanical, macroscopic, and microscopic effects, and exemplified mechanisms and define myotendinous junction healing in practice.

## 2. Materials and Methods

### 2.1. Animals

This study was conducted with 12 weeks old, 300–350 g body weight, male albino Wistar rats, randomly assigned at 6 rats/group/interval. Rats were bred in-house at the Pharmacology animal facility, School of Medicine, Zagreb, Croatia. The animal facility was registered by the Directorate of Veterinary (Reg. No: HR-POK-007). Laboratory rats were acclimated for five days and randomly assigned to their respective treatment groups. Laboratory animals were housed in polycarbonate (PC) cages under conventional laboratory conditions at 20–24 °C, relative humidity of 40–70%, and noise level 60 dB. Each cage was identified with dates, number of studies, group, dose, number, and sex of each animal. Fluorescent lighting provided illumination 12 h per day. Standard good laboratory practice (GLP) diet and fresh water were provided ad libitum. Animal care was in compliance with standard operating procedures (SOPs) of the Pharmacology animal facility, and the European Convention for the Protection of Vertebrate Animals used for Experimental and other Scientific Purposes (ETS 123).

This study was approved by the local Ethics Committee. Ethical principles of the study complied with the European Directive 010/63/E, the Law on Amendments to the Animal Protection Act (Official Gazette 37/13), the Animal Protection Act (Official Gazette 135/06), the Ordinance on the protection of animals used for scientific purposes (Official Gazette 55/13), Federation of European Laboratory Animal Science Associations (FELASA) recommendations and the recommendations of the Ethics Committee of the School of Medicine, University of Zagreb. The experiments were assessed by observers blinded as to the treatment.

### 2.2. Surgery

The surgical procedure was carried out in a blinded fashion. Under anesthesia (sodium thiopental 50 mg/kg i.p. and diazepam 6 mg/kg i.p.) with a sterile technique, the anterior side of the right knee was longitudinally incised. With precise blunt dissection, the tendon fibers were removed from the muscles, taking care not to damage the tendon-bone insertion on the patella or the muscle-bone insertion on the proximal third of the femur. It was clearly noticeable that the tendon fibers entered deep into the muscle belly and, after separation, formed a fan-shaped formation with a narrower end on the patellar insertion and a wide one on the muscular, free side. It is important that this procedure did not transect the muscles, but the tendons were completely separated from the muscles. Consequent to muscle retraction, a large defect appeared between dissected ends. The skin was closed with the running or continuous stitch (Vycril, 3.0, Ethicon Inc., Somerville, NJ, USA). In particular, rats were assessed at the end of the anesthesia period, at 2 h post-operative recovery, to ensure consistency of injury based on this expected deficit in motor function.

### 2.3. Drugs

As previously in muscle and tendon studies [19,20,21,22,27,28,31,35,36,37,46,47,51,64,65,66,67,69] medication, without carrier or peptidase inhibitor, included stable gastric pentadecapeptide BPC 157 (a partial sequence of the human gastric juice protein BPC, freely soluble in water at pH 7.0 and in saline). It was prepared as a peptide with 99% (HPLC) purity (1-des-Gly peptide was the main impurity; manufactured by Diagen, Ljubljana, Slovenia, GEPPPGKPADDAGLV, M.W. 1419) (dissolved in saline, in dose and application regimens as described before (for review see, i.e., [3,4,5,6,7,8,9,10,11,12,13,14,15,16,17]).

### 2.4. Experimental Protocol and Assessments

For functional, biomechanical, microscopical, and macroscopical assessments, medication (BPC 157 (10 µg/kg, 10 ng/kg)) was applied (i) intraperitoneally, the first application immediately after surgery, last 24 h before sacrifice. Alternatively, (ii) medication (BPC 157 (10 µg/kg, 10 ng/kg)) was given perorally in drinking water (0.16 µg/mL, 0.16 ng/mL, 12 mL/rat/day). Controls received either saline (5 mL/kg) intraperitoneally or drinking water (12 mL/rat/day) till the sacrifice, at 7, 14, 28, and 42 postoperative days.

To assess immediately following post-operative recovery, and to ensure consistency of injury based on this expected deficit in motor function, for functional assessment, at 2 h post-injury, close to the end of the anesthesia period, the initial application of regimen that was used in drinking water, BPC 157 10 µg/kg or 10 ng/kg (or water (1 mL/rat) (controls)), was given intragastrically.

For assessment of the *eNOS*, *iNOS*, *nNOS*, *COX-2* mRNA levels, and NO- and MDA-levels in the defect formed after dissecting quadriceps tendon from the quadriceps muscle, medication was BPC 157 10 ng/kg perorally in drinking water (0.16 ng/mL, 12 mL/rat/day) or drinking water (12 mL/rat/day) (controls) until the time of the sample taking from the dissected myotendinous junction before the sacrifice.

Functional tests at 1, 3, 5, 7, 10, 14, 21, 28, and 42 days, as well as tensiometry, macro/microscopic assessments at 7, 14, 28, and 42 postoperative days were performed in accordance with our muscle and tendon injuries studies [19,20,21,22,31,35,36,37,47].

### 2.5. Functional Tests

#### 2.5.1. Leg Contracture of Operated Experimental Leg (EL)

We used the procedure applied in the previous experiments’ hind limb contracture [22,31] rat held upright, as for quantification of functional recovery [22,31]. Thus, in the rats held upright, we assessed failed knee, spontaneous or forced extension one examiner pulled the back legs to achieve a maximum knee joint extension in relation to the uninjured normal leg (NL) as difference Δ, mm NL-EL. To perceive particular aspects of the injured hind limb contracture presentation, measurements were made at all 3 times to all rats while rats were awake, anesthetized, or immediately after sacrifice. The assessment was with the achieved maximum knee joint extension, marking a sign on the uninjured leg at the place reached by the thumb of the operated leg. The marked sign and top of the rat’s thumb provide the distance, which was measured by roller rules. Finally, we calculate the percentage of animals presenting operated leg contracture of more than 5% relative to the normal uninjured side [22,31].

#### 2.5.2. Walking Recovery Index WRI

Permanently tottering walk means footprint length (PL) ratio between normal (N) and experimental (E), WRI = NPL/EPL, recorded with a camera positioned at 25 cm below the 1 m long-running way (i.e., confined tunnel with transparent bottom). Controlled walking patterns were done with cameras set sideways positioned at 12 cm from the running walkway. Based on the position of the toe marker, the gait cycles were divided into a stance and a swing phase, and as a sign of the walking and function failure, we assessed the presentation of the foot sliding backward as a sudden jerk of the limb towards the back at the initiation of the swing phase [21].

#### 2.5.3. Knee Joint Angle, Ankle Joint Angle, Hip Joint Angle

Further, with cameras set sideways and the rats held upright, we take the measurement of knee extension to calculate the degree of the deficit on the operated side of a knee flexion contracture. With a permanent marker, we define the hip joint crista iliaca; trochanter, the knee joint condylus femoralis, the ankle joint lateralis malleolus, and the fifth metatarsal head. Knee joint angle was defined as an insertion between the lines extended from the hip to the knee joint, ankle joint angle as an insertion between the lines extended from the knee joint to the ankle joint, hip joint angle as an insertion between the lines extended from the trochanter to the crista iliaca [81].

#### 2.5.4. Extensor Postular Thrust EPT to Motor Function Index MFI, as Indicator of the Extent of the Function Failure

We applied the MFI determination used in the quadriceps muscle transection studies [21] as a modification of the original Koka’s description [79] used for rat functional recovery following sciatic nerve transection. MFI means the amount of weight at the moment when the rat, held upright with the hind limb extended and placed upon a digital scale precision 0.001 g, starts to bear weight on the scale, with normal (N) or experimental (E) injured leg, providing equation MFI = (NEPT − EEPT)/NEPT, and MFI maximal value of l as the worst scenario [21].

#### 2.5.5. Tensiometry

The investigation in separate groups of animals, includes measurement of ability to stretch (mm) carried out in separate experiments, immediately after sacrifice, at days 7, 14, 21, 42 by a special device Lineomat (MLW Medizinische Geräte, Chemnitz, Germany). The tests were performed also with their contralateral, healthy extremity, and the data were compared to each other, using a procedure for muscle biomechanical assessment described in detail [21].

### 2.6. Macroscopic Assessment

We assessed the differences between skinned hindmost legs separated in the hip joints, the operative, and the unoperated leg was analyzed, as the relative atrophy of the muscle by measuring the thickness of the quadriceps 2 cm proximal from the patella, normal-injured muscle diameters, mm. Likewise, we assessed the area of the defect (mm^2^) resulting from the separation of the tendon and muscle fibers. Analysis was carried out using AutoCAD and its calculation of irregular body surfaces (AutoCAD 2016 for Windows by Autodesk).

### 2.7. Microscopy

Microscopy assessment was carried out in a blinded fashion. The injured leg was skinned. Muscle was precisely separated from the bones. Then, before being embedded in paraffin wax, the muscle was fixed in buffered formalin (pH 7.4) for 24 h, and dehydrated. Longitudinal 4 µm thick sections of the muscle, including the central portion of the muscle-tendon connection, were cut, and stained with hematoxylin and eosin. Additional sequential sections were performed using Gomori and Masson trichrome and Sirius red special histochemical staining for analysis of reticulin and collagen fibers, both type 1 and type 3, one of the main classes of extracellular macromolecules of the matrix, fibrous proteins. Sirius red staining with polarized microscopy was conducted to evaluate the maturity of extracellular matrix, especially collagen type 1 secretion and production, which is the major component in tendon tissue.

Periodic acid-Schiff staining (PAS) was used for the analysis of glycogen content in muscle fibers.

### 2.8. RNA Extraction and RT-PCR in the Tendon-Muscle Junction Samples

At 7, 14, 28, and 42 postoperative days, TRIzol Reagent (Thermo Scientific, Waltham, MA, USA) was used for extraction, and spectrophotometer (NanoDrop, Thermo Scientific) for quantification at 260 nm of the tissue RNA (Table 1). According to Jazvinscak Jembrek [82] we performed the reverse transcription and semi-quantitative PCR. Briefly, total RNA (1 μg) was denatured at 65 °C for 5 min together with random hexadeoxynucleotide primers (2.5 μM). The addition of the reverse transcription buffer (Invitrogen), 0.5 mM dNTPs (Takara, Shiga, Japan), 40 U of RNase-inhibitor (Roche, USA), and 200 U of SuperScript II reverse transcriptase (Invitrogen, Waltham, MA, USA) was used for the first strand of cDNA synthesis. For DNA synthesis, we incubated the reaction mixture at 42 °C for 50 min and then heated it (70 °C, 15 min), after primer annealing (25 °C, 10 min). The sample without SuperScript II reverse transcriptase and the sample without the RNA template to test for contamination with genomic DNA was used as two negative controls for each RT reaction. EmeraldAmp MAX PCR Master Mix (Takara, Japan) with added primers (0.2 μM) was used to amplify the resulting cDNA (1:10 dilution). We performed the reactions for a determined number of cycles (in two consecutive cycles (22–23 for GAPDH and 27–28 for *eNOS*) in the logarithmic phase of PCR reaction, cDNA was amplified and analyzed). Each consisted of 95 °C for 30 s, 60 °C for 30 s and 72 °C for 1 min. The final extension was at 72 °C for 7 min. We separated the reaction products by electrophoresis on 1.5% agarose gels and visualized them by EtBr staining. ImageJ software (NIH, USA) was used for the determination of the maximal optical density. The expression of *eNOS* and *COX-2* mRNA was normalized to the reference gene GAPDH mRNA expression.

### 2.9. Oxidative Stress in the Tendon-Muscle Junction Samples

The assessment was at 7, 14, 28, and 42 postoperative days. In the tissue samples, we assessed oxidative stress by quantifying the thiobarbituric acid (TBA) reactivity as malondialdehyde equivalents (MDA). For homogenization of the tissue samples, we added trichloroacetic acid (TCA). We centrifuged the samples (3000 rpm, 5 min), and collected the supernatants. Then, we added 1% TBA and boiled the samples (95 °C, 60 min). We maintained the tubes in ice for 10 min and determined the absorbance at the wavelengths of 532 and 570 nm. We determined MDA concentration from the standard calibration curve plotted using 1,1,3,3′-tetra-ethoxy propane (TEP). We expressed the lipid peroxidation extent as the concentration of MDA, using a molar extinction coefficient for MDA of 1.56 × 105 mol/L/cm. The results are expressed in nmol/mg of protein.

### 2.10. Nitric Oxide Determination in the Tendon-Muscle Junction Samples

The assessment was at 7, 14, 28, and 42 postoperative days. Griess reaction (Griess Reagent System, Promega, USA) was used for the determination of the nitric oxide (NO) levels in the tendon-muscle junction samples. We added sulfanilamide to the homogenized tissue. Tissue was incubated, and then, we added N-(1-naphthyl)ethylenediamine dihydrochloride. The Griess reaction takes the diazotization reaction (i.e., acidified nitrite reacts with diazonium ions; then, it is coupled to N-(1-naphthyl)ethylenediamine dihydrochloride to form a chromophoric azo derivative). Sodium nitrite solution was the standard to measure the absorbance at 540 nm. The NO levels are expressed in µmol/mg protein. A commercial kit (BioRad Protein DR Assay Reagent Kit, Sigma-Aldrich, St. Louis, MO, USA) was used for the determination of the protein.

### 2.11. Statistical Analysis

The statistical analyses were with parametric one-way ANOVAs with post hoc Newman-Keuls tests and non-parametric Kruskal-Wallis tests with subsequent Mann-Whitney U tests to compare groups. The values are represented as the mean ± SD as well as the minimum, median, and maximum. The results were considered significant at *p* < 0.05.

## 3. Results

We evidenced that the stable gastric pentadecapeptide BPC 157 recovers myotendinous junction, likely with the functional recovery, muscle size recovery, and counteraction of the oxidative stress, and particular effect on *COX 2*, *nNOS*, *iNOS*, *eNOS* mRNA levels in the dissected myotendinous junction (Figure 1, Figure 2, Figure 3, Figure 4, Figure 5, Figure 6, Figure 7, Figure 8, Figure 9, Figure 10, Figure 11, Figure 12, Figure 13,Figure 14 and Figure 15). Otherwise, the injured rats remain largely disabled after careful dissecting of the quadriceps tendon from the quadriceps muscle, unable to compensate for failed crucial knee extensor.

### 3.1. Functional Recovery

Inability to compensate failed crucial knee extensor means an imminent spontaneous injured leg contracture in the control rats, which is then permanent, and equally present in all rats, in the awake or anesthetized rats, as well as immediately after sacrifice (Figure 1, Figure 2, Figure 3, Figure 4 and Figure 5). The rats maintained knee flexure, inability to achieve knee extension even upon forced extension (as confirmed by the knee joint ankle assessment, as well as assessment of the ankle joint angle, hip joint angle) (Figure 1 and Figure 2).

**Figure 1 biomedicines-09-01547-f001:**
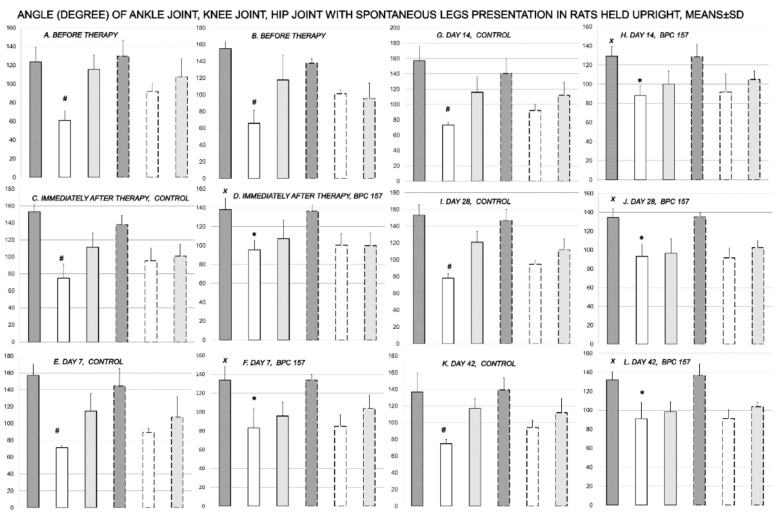
Rats, with the disabled myotendinous junction, were held upright to assess the degree of the angle of the ankle joint (dark gray bars), knee joint (white bars), and hip joint (light gray bars), and to verify spontaneous injured leg contracture (full bars) in comparison with the non-injured leg (dashed bars) after surgery, but before therapy (***A****,**B***); and subsequently, immediately after therapy application (control (***C***); BPC 157 (***D***)) (day 0), and then at day 7 (control (***E***); BPC 157 (***F***)), day 14 (control (***G***); BPC 157 (***H***)), day 28 (control (***I***); BPC 157 (***J***)) and day 42 (control (***K***); BPC 157 (***L***)); providing the course for controls *A* (before therapy)—*C* (immediately after therapy)—*E* (day 7)—*G* (day 14)—I (day 28)—*K* (day 42), and for BPC 157 *B* (before therapy)—*D* (immediately after therapy)—*F* (day 7)—*H* (day 14)—*J* (day 28)—*L* (day 42). # *p* < 0.05, at least, vs. the corresponding knee joint at non-injured leg; *x p* < 0.05, at least, vs. ankle joint in corresponding control; * *x p* < 0.05, at least, vs. knee joint in corresponding control or presentation of the knee joint in the injured leg before therapy application. Illustrative presentation of BPC 157 regimens effects includes the effect of the application of BPC 157 10 ng/kg, initial intragastric, and then in drinking water while controls received water, initially as intragastric 1 mL application, and then drinking water (12 mL/rat/day) until the sacrifice. Likewise, the same presentation was obtained with rats, which were awake, anesthetized, or immediately after sacrifice (data not specifically shown).

**Figure 2 biomedicines-09-01547-f002:**
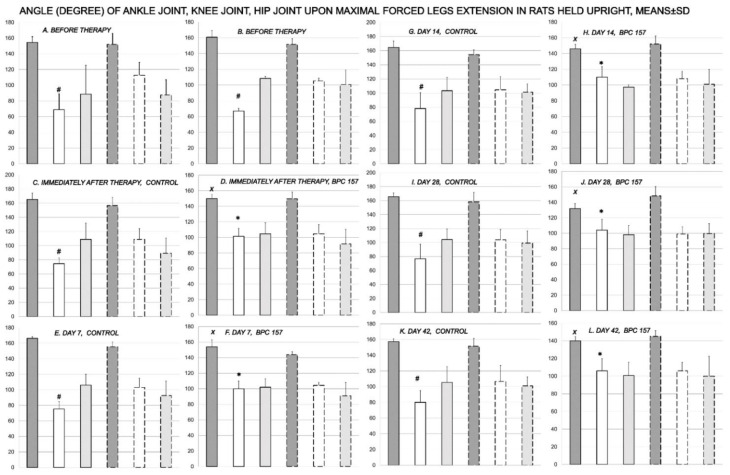
Rats, with the disabled myotendinous junction, were held upright to assess the degree of the angle of the ankle joint (dark gray bars), knee joint (white bars), and hip joint (light gray bars), and to verify injured leg contracture (full bars) in comparison with the non-injured leg (dashed bars), upon maximal legs extension after surgery, but before therapy (***A***,***B***); and subsequently, immediately after therapy application (control (***C***); BPC 157 (***D***)) (day 0), and then at day 7 (control (***E***); BPC 157 (***F***)), day 14 (control (***G***); BPC 157 (***H***)), day 28 (control (***I***); BPC 157 (***J***)) and day 42 (control (***K***); BPC 157 (***L***)); providing the course for controls *A* (before therapy)—*C* (immediately after therapy)—*E* (day 7)—*G* (day 14)—I (day 28)—*K* (day 42), and for BPC 157 *B* (before therapy)—*D* (immediately after therapy)—*F* (day 7)—*H* (day 14)—*J* (day 28)—*L* (day 42). # *p* < 0.05, at least, vs. the corresponding knee joint at non-injured leg; *x p* < 0.05, at least, vs. ankle joint in corresponding control; * *x p* < 0.05, at least, vs. knee joint in corresponding control or presentation of the knee joint in the injured leg before therapy application. Illustrative presentation of BPC 157 regimens effects includes the effect of the application of BPC 157 10 ng/kg, initial intragastric, and then in drinking water while controls received water, initially as intragastric 1 mL application, and then drinking water (12 mL/rat/day) until the sacrifice. Likewise, the same presentation was obtained with rats, which were awake, anesthetized, or immediately after sacrifice (data not specifically shown).

**Figure 3 biomedicines-09-01547-f003:**
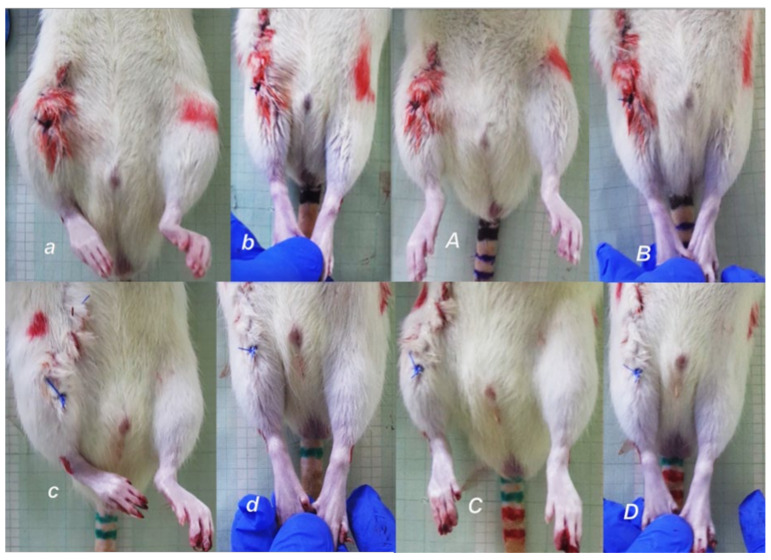
To assess immediately following post-operative recovery, and to ensure consistency of injury based on this expected deficit in motor function, at 2 h post-injury, close to the end of the anesthesia period, rats, held upright, exhibit the spontaneous injured leg contracture (***a***,***c***) that was demonstrated upon leg extension (***b***,***d***). Upon BPC 157 10 ng/kg given intragastrically (***A***,***B***), leg contracture disappears, in spontaneous presentation (***A***), or upon maximal legs extension (***B***). Contrarily, the leg contracture remains unchanged in the controls (water (1 mL/rat) given intragastrically) (***C***,***D***).

**Figure 4 biomedicines-09-01547-f004:**
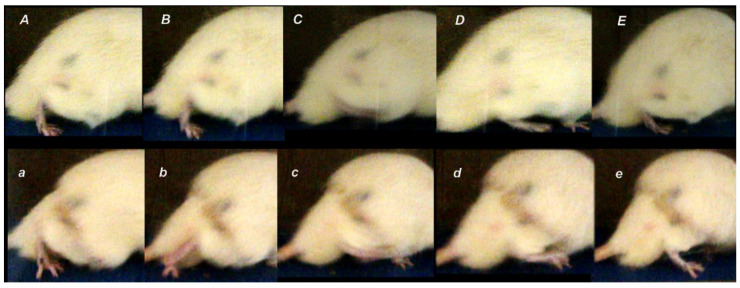
Illustrative presentation of the walking (cameras set sideways positioned) in the controls (low, ***a***–***e***) and BPC 157 treated rats (upper, ***A****–**E***). *a*, *A*. Stance phase. Control. Dorsiflexion foot, extended fingers, flexed knee, hip extension (***a***). BPC 157. Dorsiflexion foot, extended fingers, less flexed knee, hip less extension (***A***). *b, B.* Control initiates the swing phase, with prominent plantar flexion foot, flexed knee, and hip hyperextension (***b***). BPC 157 rat still in the stance phase, dorsiflexion foot, extended fingers, less flexed knee, hip extension (***B***). *c*, *C. Controls.* Swing phase. Dorsiflexion foot, flexed fingers, more flexed knee, hip flexed (***c***). *BPC 157.* Dorsiflexion foot, not flexed fingers, less flexed knee, hip flexed (***C***). *d, D. Controls.* Swing phase. Dorsiflexion foot, flexed fingers, more flexed knee, hip flexed (***d***). *BPC 157.* Dorsiflexion foot, not flexed fingers, less flexed knee, hip flexed (***D***). *e*, *E.* Stance phase. Control. Dorsiflexion foot, extended fingers, flexed knee, hip extension (***e***). BPC 157. Dorsiflexion foot, extended fingers, less flexed knee, hip less extension (***E***). Presentation of the foot sliding backward as a sudden jerk of the limb towards the back at the initiation of the swing phase (***b***) taken as an indicator of the failed walking, consistently present in control rats, but not in BPC 157 rats. Presentation at day 7 illustrative for all experiments.

**Figure 5 biomedicines-09-01547-f005:**
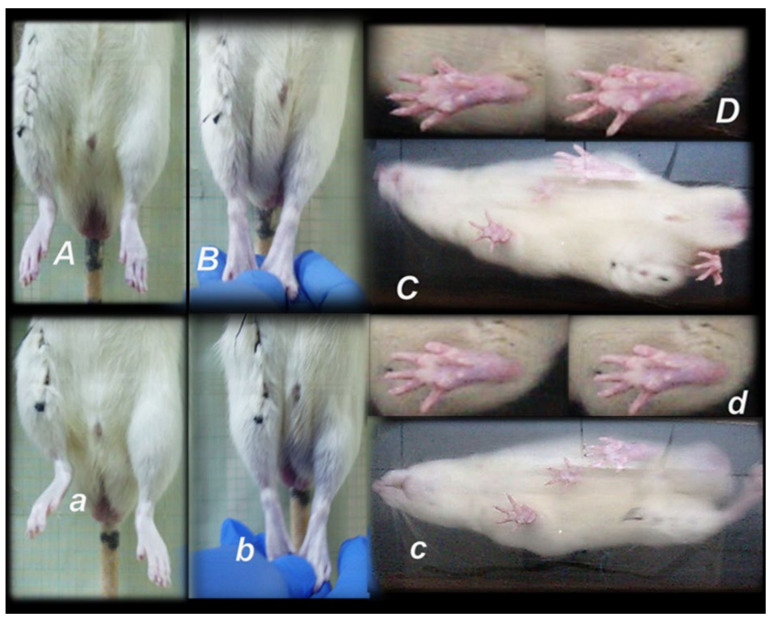
Illustrative presentation of leg contracture, day 3, (presentation (controls), and counteraction (BPC 157)) in awake rats, held upright, markedly affecting the walking presentation (camera positioned below the running way). Controls may exhibit prominent spontaneous injured leg contracture (***a***), clearly evident upon legs extension (***b***), which were completely absent in BPC 157 treated rats (***A***,***B***) and while walking (three standing point position) foot sliding backward as a sudden jerk of the limb towards the back at the initiation of the swing phase (***c***) (see also Figure 4, ***b***) and consistently less footprinting due to disturbed ankle-knee synergy (***d***), unlike presentation of the walking (***C***) and footprinting (***D***) in BPC 157 rats close to normal.

Consequently, the dissection of the muscle-tendon junction and strict knee flection result in the severe impairment of the MFI, WRI (the foot sliding backward and a sudden jerk of the limb towards the back to initiate swing phase, less footprinting—points completely lacking in the treated rats), and declined failure load (Figure 5, Figure 6, Figure 7 and Figure 8). Namely, the leg contractures disappeared in those rats having dissected quadriceps tendon from quadriceps muscle that received BPC 157, either immediately after surgery (intraperitoneal regimen), or quite later (intragastric application), and they did not have injured leg contracture during the whole experimental period (Fisher exact probability test: *p* < 0.05) (Figure 1, Figure 2, Figure 3, Figure 4 and Figure 5). As seen with the knee joint angle (as well as ankle joint angle, hip joint angle) assessment, the treated animal stop to exhibit spontaneous injured hind limb contracture, and in particular, maintains knee extension, ability to achieve an additional knee extension upon forced extension. Of note, this point is particularly evident in the rats held upright, and the mentioned injured hind limb contracture presentation (in all controls) or no presentation (in all BPC 157 rats) was equally noted when rats were awake, anesthetized, or immediately after sacrifice (Fisher exact probability test: *p* < 0.05) (Figure 1, Figure 2, Figure 3, Figure 4 and Figure 5). Consequently, the dissection of the muscle-tendon junction after and with BPC 157 therapy means a functional recovery, and as a result, these rats have much less impairment of the MFI, improved WRI. Illustratively, none of these BPC 157 rats appear with the foot sliding backward and a sudden jerk of the limb towards the back to initiate swing phase (Fisher exact probability test: *p* < 0.05)); the improved are MFI and WRI, along with biomechanic improvement, failure load close to the non-injured leg (Figure 5, Figure 6, Figure 7 and Figure 8).

**Figure 6 biomedicines-09-01547-f006:**
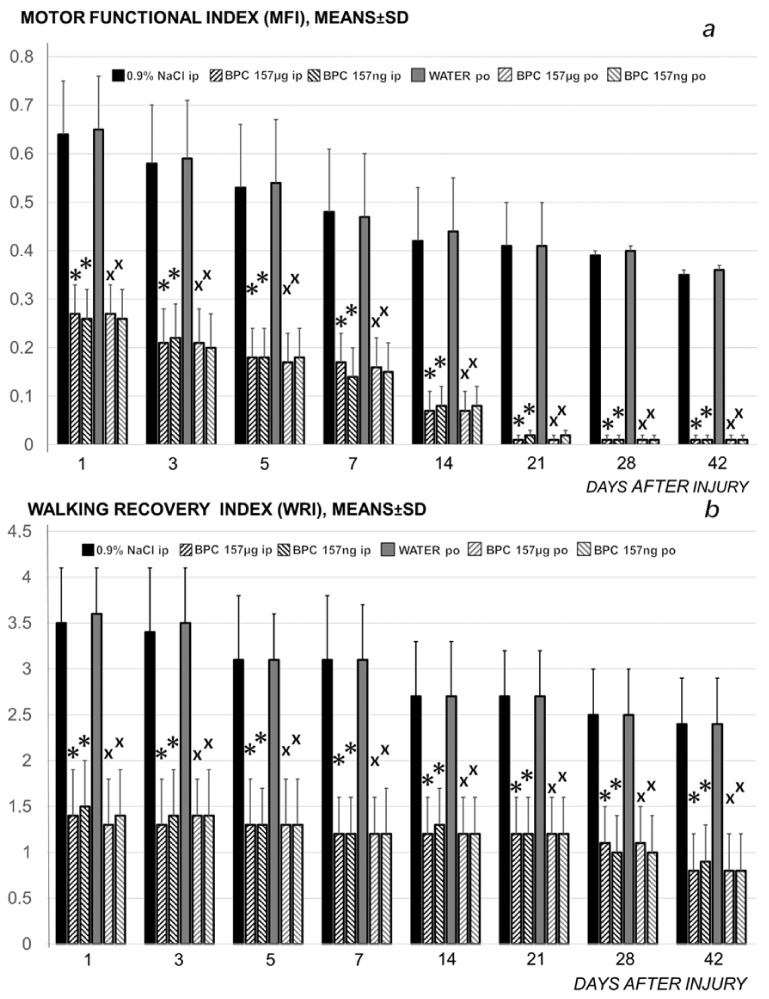
Rats with the disabled myotendinous junction, functional tests (motor functional index (MFI) (upper) (***a***), walking recovery index (WRI), mm (lower) (***b***), means ± SD) at 1, 3, 5, 7, 10, 14, 21, 28 and 42 days. Higher MFI and WRI values signify higher failure; higher MFI signifies more muscle function failure; higher WRI signifies more walking function failure. Medication (BPC 157 (10 µg/kg, 10 ng/kg)) was applied either intraperitoneally (BPC 157 µg i.p.; BPC 157 ng i.p.), first application immediately after surgery, last 24 h before sacrifice, or perorally in drinking water (0.16 µg/mL, 0.16 ng/mL, 12 mL/rat/day) (BPC 157µg po; BPC 157 ng po) while controls received either saline (5 mL/kg) intraperitoneally or drinking water (12 mL/rat/day) till the sacrifice, at 7, 14, 28 and 42 postoperative days.* *p* < 0.05, at least, vs. control(ip); x *p* < 0.05, at least, vs. control (po).

**Figure 7 biomedicines-09-01547-f007:**
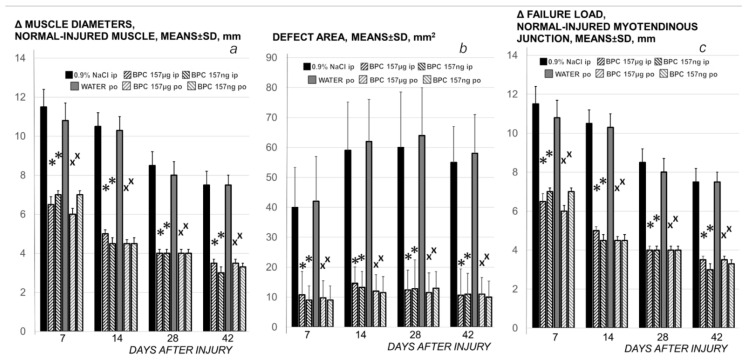
Rats with the disabled myotendinous junction, muscle diameters (as Δ between normal and injured muscle) (***a***); defect area, mm^2^ (***b***), and biomechanics (***c***) assessment, means ± SD at 1, 3, 5, 7, 10, 14, 21, 28 and 42 days. Assessments were macroscopic (the area of the defect resulting from the separation of the tendon and muscle fibers, mm^2^), microscopic (normal-injured muscle diameters, mm); tensiometry (special device Lineomat, MLW Medizinische Geräte, Chemnitz, Germany, mm, means ± SD), performed in accordance with our muscle and tendon injuries studies [19,20,21,22,31,35,36,37,47]. Medication (BPC 157 (10 µg/kg, 10 ng/kg)) was applied either intraperitoneally (BPC 157 µg ip; BPC 157 ng ip), first application immediately after surgery, last 24 h before sacrifice, or perorally in drinking water (0.16 µg/mL, 0.16 ng/mL, 12 mL/rat/day) (BPC 157 µg po; BPC 157 ng po) while controls received either saline (5 mL/kg), intraperitoneally or drinking water (12 mL/rat/day) till the sacrifice, at 7, 14, 28 and 42 postoperative days.* *p* < 0.05, at least, vs. control (ip); x *p* < 0.05, at least, vs. control (po).

**Figure 8 biomedicines-09-01547-f008:**
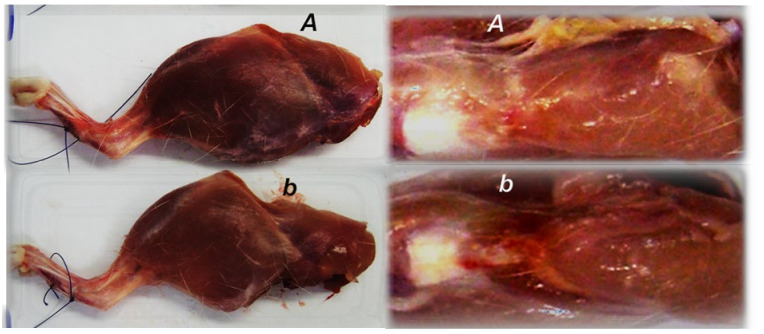
Illustrative presentation of the skinned injured rat leg and defect presentation at day 28 after myotendinous injury induction, in control (low, ***b***) and BPC 157 treated (***A***) rats. Muscle atrophy and persistent defect in controls in contrast to presentation close to normal and reestablished myotendinous junction in BPC 157 treated rat.

### 3.2. Muscle Size Recovery

In relation to continuous strong distinction in the muscle diameters between the normal and the injured leg in the controls and the BPC 157 treated rats (Figure 7) and skinned injured leg gross presentation (Figure 8), as a particular notation are the findings that control rats exhibit areas of smaller muscle fibers in diameter at 28th day, with progressed muscle fiber atrophy at 42nd day (Figure 13).

During this atrophy process, a diversity of muscle fiber types seems to vanish according to PAS staining results. While at the 28th day, a diversity of muscle fiber types still persists resulting in different PAS staining (PAS intensity staining + and +++), at 42nd day this diversity vanished, only residual muscle fibers with low PAS intensity staining were observed (Figure 14a,b). In contrast, BPC 157 group had a well-preserved thickness of muscle fibers through all experiments (Figure 13), and diversity of muscle fiber types at PAS staining respectively (Figure 14c,d).

Likely with cause-consequence relation with functional assessments (Figure 1, Figure 2, Figure 3, Figure 4, Figure 5, Figure 6 and Figure 7), in controls, both macroscopic and microscopic assessment substantiates that the regular healing of the dissected muscle-tendon junction fails to approach the original structure (Figure 8, Figure 9 and Figure 10). Grossly, the defect is constantly present, and as well muscle atrophy, regularly progressive. Contrarily, BPC 157 counteracts muscle and brings muscle presentation close to the normal, and rats’ BPC 157 therapy exhibit a smaller defect, and finally defect completely disappears as completely filled (Figure 8).

**Figure 9 biomedicines-09-01547-f009:**
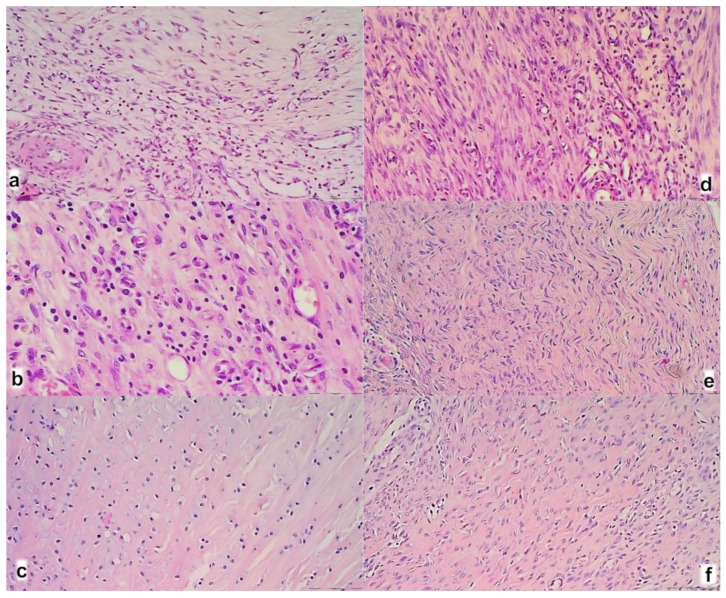
A myotendinous junction area in control (**a**–**c**) and BPC 157 treated rats (**d**–**f**); at day 7 (**a**,**b**), day 14 (**b**,**e**), and day 28 (**c**,**f**) after myotendinous injury. (HE; magnification ×200).

The control group showed at 7th day (Figure 9a) prominent edema, infiltration of inflammatory cells, and reduction of capillaries with survived arterioles of the myotendinous junction; at 14th day (Figure 9b) a persistent edema and infiltration of inflammatory cells, delayed and discrete increase of vascularity of MTJ; at 28th day (Figure 9c) no edema and inflammatory cells with vanishing revascularization in MTJ area. A muscle fiber atrophy with cylindrical or completely atrophied fibers randomly distributed within myotendinous junction area. BPC 157 treated rats showed at 7th day (Figure 9d) mild edema, infiltration of inflammatory cells, and significant vascularity of myotendinous junction with penetrating capillaries; at 14th day (Figure 9e) no edema and inflammatory cells, completely vanished revascularization of MTJ, only well orientated dense connective tissue; at 24th day (Figure 9f) a well orientated dense connective tissue, no randomly distributed muscular fibers were found within MTJ area. Morphologic features of the MTJ area indicate that BPC 157 therapy favors vascular density as well as reconstruction and orientation of reticulin and collagen fibers.

**Figure 10 biomedicines-09-01547-f010:**
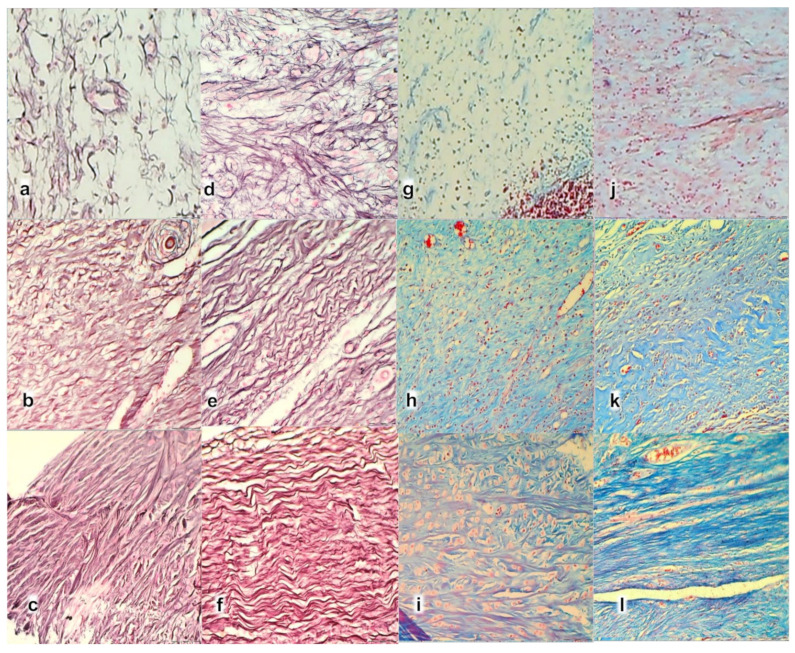
Histochemical staining of myotendinous junction area after myotendinous injury in control (**a**–**c**,**g**–**i**) and BPC 157 treated rats (**d**–**f**,**j**–**l**); at day 7 (**a**,**d**,**g**,**j**), day 14 (**b**,**e**,**h**,**k**), and day 28 (**c**,**f**,**i**,**l**), (Gomori staining (**a**–**f**); Masson trichrome staining (**g**–**l**), magnification ×200).

Gomori staining in the control group showed at 7th day (Figure 10a) disintegrated reticulin fiber network; at 14th day (Figure 10b) a discrete proliferation of fibroblast and production of reticulin and collagen fibers making loose connective tissue with mesh-like fibers, areolar tissue. All fibers were gray, implying reticulin and collagen type 3 fibers; at 28th day (Figure 10c) a moderate proliferation of fibroblasts and fibers, and fibers maturation with the suboptimal orientation of reticulin fibers to the long axes of the myofibers close to the myotendinous junction. BPC 157 group showed at 7th day (Figure 10d) prominent proliferation of fibroblast with the synthesis of reticulin and collagen fibers. All fibers were gray, implying reticulin and collagen type 3 fibers; at 14th day (Figure 10e) prominent and well-orientated fibroblast and collagen fibers proliferation. A prominent collagen maturation with numerous black fibers of collagen type 1; at 28th day (Figure 10f) fibroblast and fibers proliferation and maturation with optimal orientation to the long axes of the myofibers close to the myotendinous junction with lesser number of fibroblasts, and more amount of reticulin and collagen fibers.

Masson trichrome staining in the control group showed at 7th day (Figure 10g) no proliferation of collagen fibers; at 14th day (Figure 10h) production of collagen fibers type 3 making loose connective tissue with mesh-like fibers; at 28th day (Figure 10i) a moderate proliferation and maturation of collagen fibers type 1 with insufficient orientation to the long axes of the myofibers close to the myotendinous junction. Atrophy of muscle fibers at myotendinous junction occurred like cylindrical or completely atrophied, randomly distributed within myotendinous junction area. BPC 157 group showed at 7th day (Figure 10j) proliferation of collagen type 3 fibers; at 14th day (Figure 10k) prominent proliferation and maturation of collagen fibers type 1; at 28th day (Figure 10l) a consistent prominent proliferation of collagen fibers type 1. No randomly distributed muscular fibers were found within the MTJ area.

The intensity of Sirius red staining was significantly higher in treated animals than in control groups at all three-time intervals (Figure 11). Polarized light microscopy image showing the maturation of the newly formed collagen fibrils in BPC 157 treated animals in contrast with the control group, with minimal or no production fibrils in the control group at day 7 (Figure 11a), indicating significantly increased production of collagen type 1 in BPC 157 rats.

Likewise, along with both functional and gross clinical assessments, microscopic assessment demonstrates that BPC 157 therapy favors vascular density as well as reconstruction and orientation of reticulin and collagen fibers. Specifically, at day 7, there is an infiltration of inflammatory cells while controls presented prominent edema, disintegrated reticulin fiber network, and no proliferation of collagen fibers (Figure 9 and Figure 10), and reduction of capillaries with survived arterioles of the myotendinous junction. In BPC 157 rats, there is the increased vascularity of the myotendinous junction with penetrating capillaries, and proliferation of fibroblast was pronounced with the synthesis of reticulin and collagen fibers (Figure 9d and Figure 10). At Gomori staining all fibers were gray, implying reticulin and collagen type 3 fibers (Figure 10), and positive collagen type 1 fibers were obtained using polarized microscopy Sirus red staining (Figure 11).

On day 14, there is still an infiltration of inflammatory cells in controls, presented moderate edema which disappears not before the day 28 (Figure 9), sustainable, but delayed discrete increase of vascularity with the discrete proliferation of fibroblast and production of reticulin and collagen fibers making loose connective tissue with mesh-like fibers, areolar tissue. Fibers are far enough apart to leave ample open space for interstitial fluid in between. It is strong enough to bind different tissue types together, yet soft enough to provide flexibility and cushioning (Figure 9 and Figure 10). Sparse and not well-organized collagen type 1 fibers beginning on the 14th day were observed in the control (Figure 11).

At the same time, BPC 157 rats show that edema completely vanished. They present revascularization of the myotendinous junction, prominent fibroblast proliferation with reticulin, and collagen fibers synthesis that was all well orientated (Figure 9 and Figure 10). There is a maturation of collagen fibers as easily seen at Gomori and Sirius red stainings. The appearance of black fibers within gray fibers implies collagen type 3 fibers within reticulin fibers and the myotendinous junction is reestablished according to morphological features (Figure 9 and Figure 10), and the appearance of abundant and well-orientated collagen type 1 fibers was found (Figure 11). This dense collagen tissue is both strong and flexible enough to bind different tissue types together.

**Figure 11 biomedicines-09-01547-f011:**
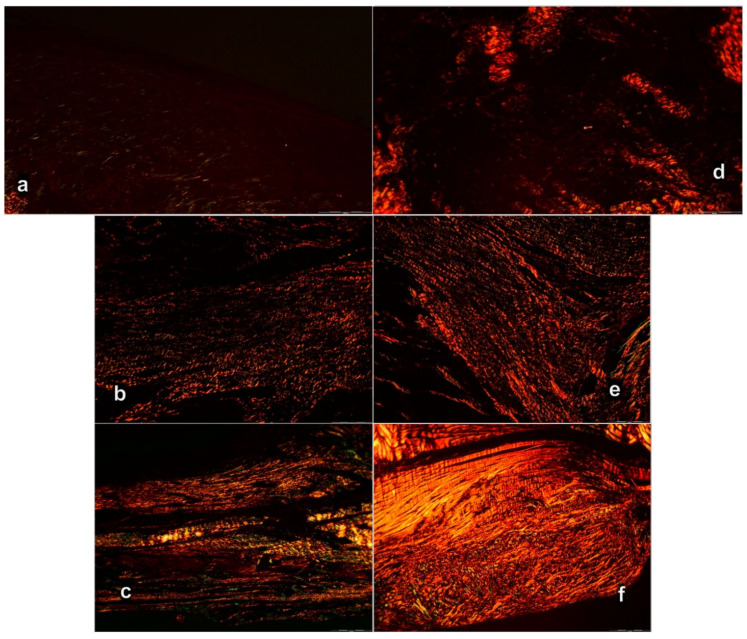
Sirius red histochemical staining with polarized microscopy of myotendinous junction area after myotendinous injury in control (**a**–**c**) and BPC 157 treated rats (**d**–**f**). Day 7 (**a**,**b**), day 14 (**b**,**e**), and day 28 (**c**,**f**) (Sirius red staining, magnification ×200).

A significant difference between production and maturation of collagen at day 28 (as well as at day 42), in control group revascularization of the myotendinous junction, vanished, the proliferation of fibroblasts and fibers, as well as fibers maturation, was present but not to the extent seen in BPC 157 treated rats (Figure 10 and Figure 11). Fibers orientation was not optimal due to the long axes of the myofibers close to the myotendinous junction (Figure 9). Also, due to clinical contraction and spared use of the injured leg, muscle fiber atrophy occurred (Figure 13).

Morphologically, the finger-like muscle cell endings become shallow and cylindrical or completely atrophied (Figure 12, dashed border). Simultaneously, BPC 157 rats exhibit prominent proliferation of fibroblasts and fibers as well as fibers maturation, providing tenable fibroblast and fibers proliferation with their optimal orientation due to the long axes of the myofibers close to the myotendinous junction with lesser number of fibroblasts, and more amount of reticulin and collagen fibers. No muscle fiber atrophy was noted. Morphologically, the finger-like muscle cell endings were preserved (Figure 12, dashed border).

**Figure 12 biomedicines-09-01547-f012:**
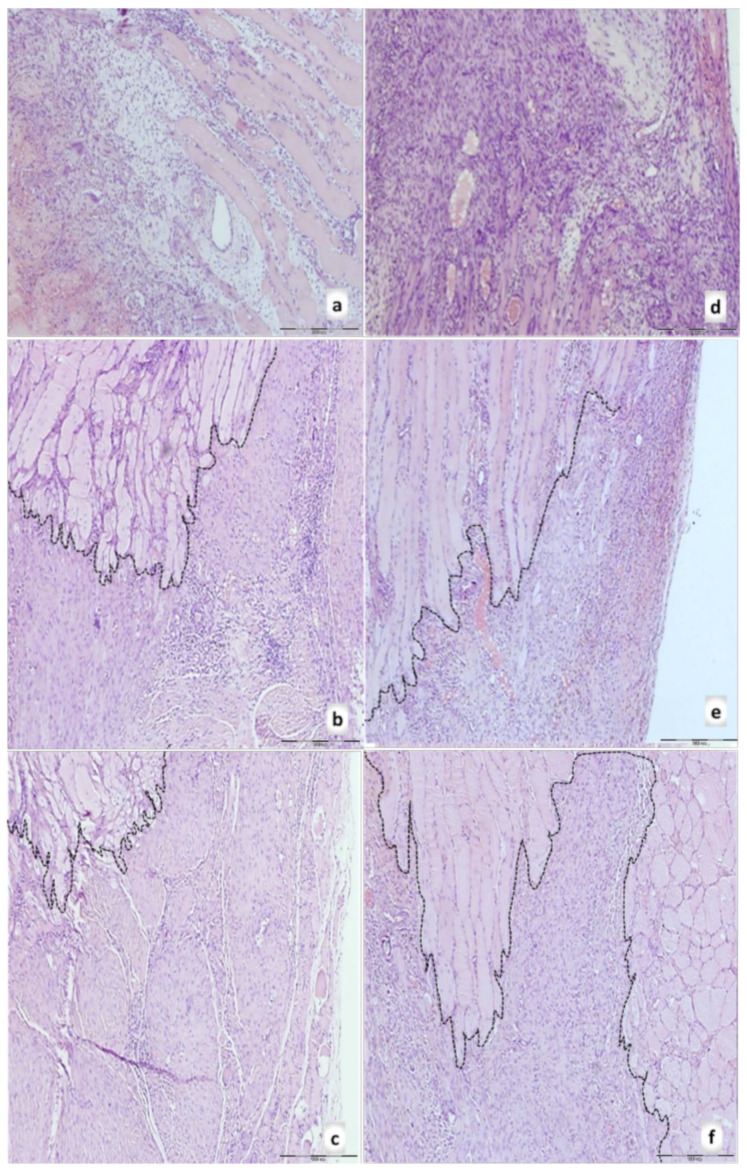
Rats with the disabled myotendinous junction, microscopic presentation of the muscle-tendon junction in control (**a**–**c**) and BPC 157 (**d**–**f**) rats at day 7 (upper), day 14 (middle), and day 28 (lower) of the myotendinous junction injury. Dashed border of the muscle cell endings (HE staining; magnification ×100). Control group: prominent edema, infiltration of inflammatory cells, and reduction of capillaries with survived arterioles of the MTJ (7th day) (**a**); a delayed and discrete increase of vascularity of myotendinous junction (14th day) (**b**); a muscle fiber atrophy with the shallow and cylindrical muscle cell endings or completely atrophied in myotendinous junction area, most prominent at 28th day (see dash border with larger radius and smaller curvature between muscle and connective tissue fibers at 14th and 28th day; each fiber with the cylindrical muscle cell endings (**c**). BPC 157 group: a prominent proliferation of fibroblast with the synthesis of reticulin and collagen fibers of the myotendinous junction with significant vascularity and penetrating capillaries, mild edema, and infiltration of inflammatory cells (7th day (**d**)); no edema and inflammatory cells, completely vanished revascularization of the myotendinous junction, only well orientated dense connective tissue (14th day) (**e**); a well orientated dense connective tissue, no randomly distributed muscular fibers were found within myotendinous junction area (28th day) (**f**). No muscle fiber atrophy within the myotendinous area was found (**f**). Morphologically, the finger-like muscle cell endings were very well preserved, sharply, and regularly distributed within the myotendinous area (see dash border with a smaller radius and larger curvature between muscle and connective tissue fibers at 14th and 28th day; each muscle fiber with the finger-like cell endings).

**Figure 13 biomedicines-09-01547-f013:**
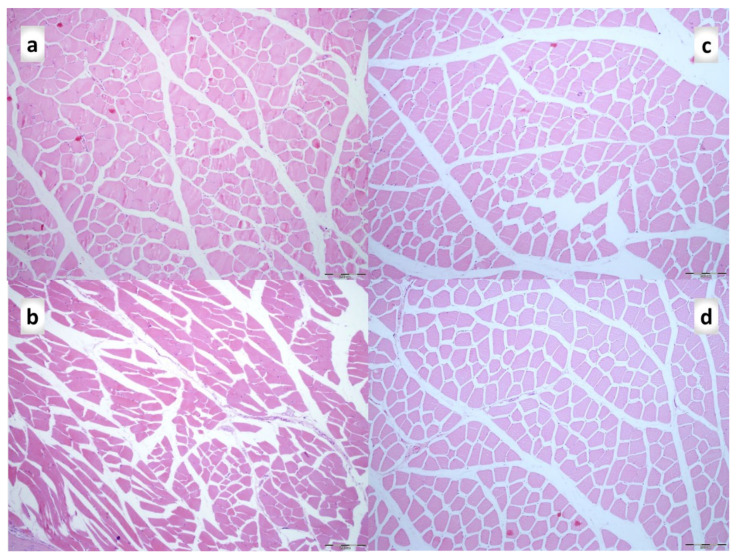
Hematoxylin and eosin-stained cross-sections of the quadriceps muscle. (**a**). Control rats at day 28 (**b**). Control rats at day 42. Smaller fiber areas can be appreciated at day 28, and more diffuse smaller fiber areas found at day 42 compared to the BPC 157 group at day 28 (**c**) and day 42 (**d**) of injury. Magnification ×100. Scale bar is 200 μm.

On the 42nd day, morphologic features were constant in both groups compared to those on the 28th day. In the control group, a proliferation of fibroblasts and fibers as well as fibers maturation was present but still not so prominent compared with the BPC157 group, and their orientation was still not optimal due to the long axes of the myofibers close to the myotendinous junction. Continuous muscle atrophy persisted.

In BPC 157 rats’ fibers proliferation with optimal orientation due to the long axes of the myofibers close to the myotendinous junction with lesser number of fibroblasts, and more amount of reticulin and collagen fibers persisted, with no muscle fiber atrophy.

A loss of muscle fibers diversity results in monotonous PAS staining with low intensity of staining was found in control rats on the 42nd day, compared to BPC 157 treated rats in which diversity of muscle fibers remained (Figure 14).

**Figure 14 biomedicines-09-01547-f014:**
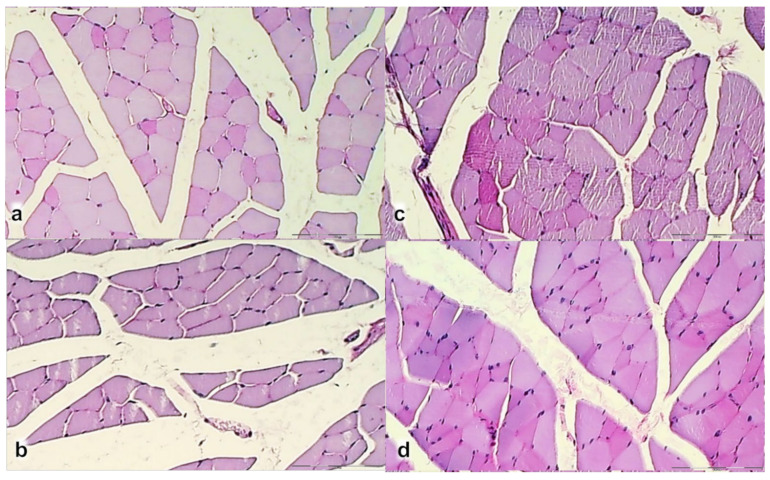
Periodic acid-Schiff stained cross-sections of the quadriceps muscle. (**a**). Control rats at day 28. The diversity of muscle fibers results in different PAS staining (PAS intensity staining + and +++). (**b**). Control rats at day 42. Loss of muscle fibers diversity results in monotonous PAS staining (PAS intensity staining +). BPC 157 rats at day 28 (**c**) and day 42 (**d**). The diversity of muscle fibers results in different PAS staining (PAS intensity staining + and +++). Magnification ×200.

### 3.3. Oxidative Stress, and Particular Effect on *COX 2*, *nNOS*, *iNOS*, *eNOS* mRNA Levels

#### 3.3.1. MDA, NO-Tissue Levels

Considering the oxidative stress in rats with disabled muscle-tendon junction, as seen with the increased MDA values during the whole experiment (in particular on day 7 and day 14), BPC 157 opposes oxidative stress, since BPC 157 maintains normal MDA values and counteracts the increased MDA values after dissection in the muscle-tendon juncture. BPC 157 decreased NO-tissue levels (Figure 15). Likely, these may be related to the NO-values course. Note, in the controls, the increased NO-tissue value occur during the whole experiment (in particular on day 7, day 14, and especially on day 28). In most periods, despite the ongoing increase that would regularly occur in controls, BPC 157 maintains normal NO-values and markedly opposes the NO-peak at day 28. This effect seems to be related to disease conditions, since, in the healthy rats, BPC 157 had no effect.

#### 3.3.2. *COX 2*, *nNOS*, *iNOS*, *eNOS* mRNA Levels

During the experiment, both *eNOS* and *COX 2* mRNA levels are increased in the disabled myotendinous junction, but with distinction. *eNOS* increase occurs during the earlier periods, *COX 2* occurs during all periods. BPC 157 has a particular effect, as it was seen with its effect on NO- and MDA-values in the myotendinous junction. Namely, BPC 157 increases *eNOS* mRNA levels and decreases *COX 2* mRNA levels (Figure 15). Again, this effect seems to be related to disease conditions, since, in the healthy rats, BPC 157 had no effect. The level of *iNOS* and *nNOS* was below the method threshold (not presented).

**Figure 15 biomedicines-09-01547-f015:**
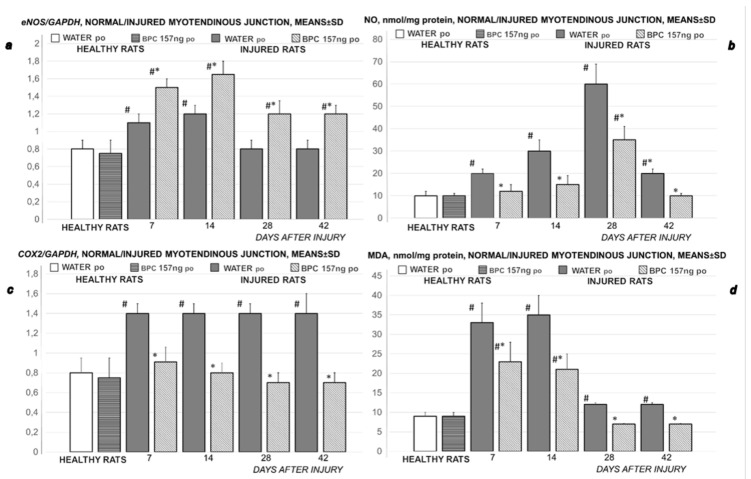
Rats with the disabled myotendinous junction, expression of *eNOS* mRNA (***a***), NO-level (***b***), expression of *COX-2* mRNA (***c***) and MDA-level in the disabled myotendinous junction (***d***), means ± SD, at 7, 14, 28, and 42 postoperative days. Medication (BPC 157 10 ng/kg) was applied perorally in drinking water (0.16 ng/mL, 12 mL/rat/day) while controls received drinking water (12 mL/rat/day) till the sacrifice, at 7, 14, 28 and 42 postoperative days.

Concluding, final evidence belongs to the consistent effect of the two distinctive regimens, given either once time daily, as intraperitoneal administration, or perorally, continuously in the drinking water (providing also the evident effect of once time initial intragastric administration), and application of the two different dose range. Together, these may suggest easy applicability, and likely, practical value.

## 4. Discussion

Regularly, the injured rats definitely remain largely disabled after careful dissecting of quadriceps tendon from the quadriceps muscle, unable to compensate failed crucial knee extensor. Contrarily, the stable gastric pentadecapeptide BPC 157 recovers myotendinous junction. Likely, since with the BPC 157 therapy [19,20,21,22,31,35,36,37,47], the muscle and tendon healing [19,20,21,22,31,35,36,37,47] goes to the myotendinous junction healing, the evidenced osteotendinous junction successfully restored [19,22] may verify a similar restoration of the other junction as well. Together, this brings the myotendinous junction restoration by BPC 157 as the particular healing course, which is obviously a specific and valuable one. This ascertains an essential process not hampered with the initial tendon healing process as similar to fracture healing [83], nor ossicles formation in other tissues as it is with the application of the bone morphogenetic proteins (BMPs) [84,85,86].

Thus, with the myotendinous junction, with cause-consequence relations, the positive outcome reestablished tendon-muscle continuity supports the functional, biomechanical, microscopical, and macroscopical findings as well as *eNOS* and *COX-2* mRNA levels, and NO- and MDA-levels in the defect formed after dissecting quadriceps tendon from the quadriceps muscle. The consistent effects of the once a day intraperitoneal regimen, or per-oral regimen, continuously in drinking water, support that BPC 157 tendon healing [18,19,20,21,22] and muscle healing [21,35,36,37], noted when either tendon or muscle was transected or detached [18,19,20,21,22], may render to the myotendinous junction healing. These mean the native peptide therapy, and the myotendinous junction recovery along with the described beneficial effect in the muscle [21,35,36,37] and tendon healing [18,19,20,22], unmistakably attributed all of its effects (i.e., functional, biomechanical, macroscopic, and microscopic effects and exemplified mechanism(s)) to define myotendinous junction healing in practice. There, the functional recovery, muscle size recovery, and oxidative stress, may be particularly illustrative.

Spontaneously, functional recovery does not occur. The failed crucial knee extensor perceives no spontaneous recovery of the severe functional deficit after careful dissecting of the quadriceps tendon from the quadriceps muscle. The supportive evidence goes for both bipedal and quadrupedal rat positions. Rat held upright follows the bipedal motion of quiet standing [80], approaches the human body connected quasilinearly while the rat’s body segments are flexed [80]. To this point goes the relevance of the findings that as in the previous muscle and ligament injuries studies [19,20,21,22,31,35,36,37,47], the damaged animal exhibits spontaneous injured hind limb contracture, in particular, maintained knee flexure, inability to achieve knee extension even upon forced extension. Thus, the relevant consequence is the high extent of the loss of the muscle strength MFI as the notation of the muscle failure, and thereby high MFI. The profoundly failed muscle strength goes with the high initial MFI-values, close to the values indicative for the worst outcome from the normal healthy values, and remain high until the end of the experimental period. Further, failed crucial knee extensor and thereby prevalent remained other muscles function i.e., knee flexion, plantar flexion disrupts the ankle-knee synergy in the rat quadrupedal walking [81]. This induces an opposite combined effect, quite immobile knee joint flexion-the more ankle joint extension. And thereby, apparently less foot length printing, less balanced posture i.e., quadrupedal standing requires control of the balance of the load for each leg [81], the foot sliding backward and a sudden jerk of the limb towards the back to initiate swing phase. Consequently, together, this impairs the use of the injured limb, thereby definitively deteriorating the healing.

Vice versa, for the BPC 157 therapy beneficial effect, the essential evidence is the noted functional recovery i.e., recovered failed crucial knee extensor, in both bipedal recovered muscle strength small to negligible MFI; absent injured leg contracture as in the previous ligament and muscle injury studies [19,20,21,22,31,35,36,37,47] and quadrupedal improved walking assay. Likely, this reflects the improved rat myotendinous junction function ability and biomechanics, which directly reflect ongoing healing processes, and thereby, the improvement, whatever mechanism. Note that after myotendinous injury controls exhibit not only muscle fibers atrophy, but also changes in distribution and vanishing of the muscle fiber type’s diversity. Avoiding all of these changes with BPC 157 therapy may be the result of the innate reversal of the delayed and failed recovery of the myotendinous junction after injury leading to an important role in the transmission of myotendinous junction muscle fiber force to the endomysium or adjacent muscle fibers. Indicatively, the noted functional recovery is significant. The small loss of the muscle strength, small MFI goes to the achieved normal strength negligible MFI, the treated animal does not exhibit spontaneous injured hind limb contracture, and in particular, maintains knee extension, ability to achieve an additional knee extension upon forced extension. Likewise, recovered failed crucial knee extensor quite preserves walking patterns and the ankle-knee synergy in the rat having to dissect the quadriceps tendon from the quadriceps muscle. Thus, receiving BPC 157 therapy means that the treated rat exhibits a smaller defect, and finally defect completely disappears as it is filled. Such outcome corresponds to the transected quadriceps muscle findings grossly, the stumps connected; subsequently, increasing atrophy progress markedly attenuated; and finally, muscle fully approaches to the presentation of normal non-injured muscle [21]. Likewise, it corresponds to the transected or detached Achille’s tendon unlike significant gap between the tendon edge and bone with a clear stump, the BPC 157-treated rat leg shows no defect between the tendon stump and calcaneal bone and the edge of the tendon stump cannot be recognized sharply with the connective tissue matrix [19,20,22].

Furthermore, providing that research on skeletal muscle injury and regeneration highlights the crucial role of nerve-muscle interaction in the restoration of innervation during that process [87,88], with the cause-consequence relations, this may be the initial effect of BPC 157 on the neuromuscular junction integrity [65], shown when confronted with neuromuscular blocker succinylcholine. With an immediate effect i.e., succinylcholine injected into a rat (anterior tibial) muscle [65], BPC 157 given before intramuscular succinylcholine, either intraperitoneally or per-orally in drinking water or intraperitoneally shortly after intramuscular succinylcholine, thereby protocol [65] alike that used in the present study counteracts the acute systemic muscle disability and sustained local paralytic effect consequent to neuromuscular blockade. Furthermore, in rats that received BPC 157 before succinylcholine application, the injected leg contracture did not appear, or, the leg contractures disappeared in those rats that received BPC 157 after administration of succinylcholine. Of note, it was counteraction of the leg contracture, which was assessed immediately after the intramuscular injection of succinylcholine, after the cessation of overt symptoms of muscle involvement, and then at the subsequent following days. Next to counteracting paralysis and immobility, all the BPC 157 regimens were effective against consequent muscle damage, hyperkalemia, and arrhythmias [65], it is likely that BPC 157 might also counteract the initial event. Likewise, the effect of BPC 157 at 24 h following rat aortic termino-terminal anastomosis is a rapid effect [47]. Given at that point, intraperitoneally, within 3 min post-application interval, the pentadecapeptide BPC 157 rapidly recovered the function of lower limbs and muscle strength while no cloth could be seen in those rats at the anastomosis site [47]. Thus, although this BPC 157’s beneficial point is not directly investigated in the present study, the maintenance of the neuromuscular junction may be essential evidence, since skeletal muscle regeneration is functionally successful only if the myofibres that regenerate from injury also become effectively innervated through the establishment of neuromuscular junctions [87,88]. Likewise, reestablished blood flow may certainly contribute to the rapid recovery effect noted, BPC 157 directly protects endothelium [47,56], alleviates the peripheral vascular occlusion disturbances [49,50,51,52,53,54,55,56,57,58,59,60], rapidly activating alternative bypassing pathway [49,50,51,52,53,54,55,56,57,58,59,60].

Finally, as an analogy for the rapid beneficial effect noted in the rats with myotendinous junction injury, goes the rapid counteraction of the various arrhythmias with BPC 157 therapy, both rapid prevention and rapid reversal [63,65,68,89,90,91]. Consequently, improved function (here, advanced WRI and EPT/MFI, lacking leg contracture) facilitates the use of the injured limb (thereby, healing) even before the completion of the repair process. Therefore, there is a prominent improvement at both early (post-dissection healing promptly induced) and late intervals (the healing maintained), and improved biomechanics (failure load) since the very beginning. And thereby, as assessed also in the recovery course in the rats with the injured/transected tendon, ligament, or muscle [19,20,21,22,31,35,36,37,47], there is with the same therapy background the counteraction of the “leg contracture” as a reliable, reproducible, and sensitive tool. Likewise, in the rats that underwent dissecting of quadriceps tendon from the quadriceps muscle, “leg contracture”, present (in controls) or absent (in BPC 157-rats), may reflect during the experiment the persistent inability or recovered ability to achieve active knee extension motor function. Post-sacrifice assessment would eliminate as prime cause the muscular weakness, joint distension or joint stiffness, and pain, “leg contracture”, present (controls) or absent (BPC 157-rats), reflects the posture phenotype related to the continued disconnection or recovered connection of the muscle-tendon unit.

Thereby, as a cause-consequence relation appears the counteracted muscle atrophy, no more inflammatory infiltrate in the BPC 157 group, well-oriented recovered tissue of myotendinous junction in BPC 157 treated rats at the 28 days and 42 days. Initially, on day 7, BPC 157 rats had only small edema and inflammatory infiltrate, the connective tissue in more appropriate structure with only a small scar at the 14 days that would not disrupt the orientation as well as continuity of the muscle-tendon junction. As the osteotendinous junction is restored (i.e., tendon attaches to the bone with an intervening zone of fibrocartilage (direct type) that requests synthesis of strong collagen fibers of type I) [22], goes the other junction, the myotendinous junction, restoration. Type III collagen is also cleared promptly and substituted with collagen type I fibers starting at the 14th day of treatment (i.e., improved load to failure correlate with the advanced collagen type I distribution) (Figure 10e,f,k,l and Figure 11e,f). A synthesis of collagen type I starts before, at the 7th day of treatment (Figure 11d), with consequent optimal orientation to the tendon long axis at 14th day of treatment, as consistent evidence based on the specificity of the used methods for collagen staining. In addition to the tendon [18,19,20,22] and muscle [21,35,36,37] studies, that finding was also seen in thermally injured animals that received BPC 157 therapy [40,42,44].

Having ascertained beneficial effect, further support goes with mRNA expression *Nos3, Cox2* studies, and NO- and MDA-level assessment in the myotendinous junction. Likely, provided throughout all of the intervals, compared with the controls, consistently elevated (*Nos3*) and decreased (*Cox2*), decreased NO- and MDA- level, may show a way how BPC 157 may act throughout NO- and prostaglandins- systems and counteract oxidative stress. Possibly, with respect to underlying disease (which was markedly counteracted in BPC 157 rats), more activity of *eNOS* and less COX2 may be an own effect of BPC 157 [15]. In particular, this may be substituting NO-effect [15] (and thereby, lesser NO-tissue values), additive and/or synergistic effect (NO fully implemented in the healing, and thereby, in relation to the NO-significance in the muscle and tendon healing [39,77], achieved myotendinous recovery) related to the own activity of BPC 157 administration in the rats having myotendinous lesions. Likewise, less COX2 may reflect its particular anti-inflammatory effects, like that noted in the counteraction of the cachexia induced by tumors in mice [4], muscle wasting, deranged muscle proliferation and myogenesis, an increase of the proinflammatory cytokines (i.e., IL-6, TNF-α) and the changes in the expression of FoxO3a, p-AKT, p-mTOR, and P-GSK-3β. Interestingly, this anti-inflammatory effect in the muscle and tendon healing, and myotendinous junction recovery [18,19,20,21,22], is along with BPC 157 anti-inflammatory effect in other models [14], counteracted adjuvant arthritis [92], capsaicin-toxicity [93], and goes with counteraction of the NSAIDs side effects [14]. This was associated with its function as a stabilizer of cellular junction [5], leading to significantly mitigated indomethacin-induced leaky gut syndrome, via increasing tight junction protein ZO-1 expression, and transepithelial resistance [5]. Likewise, there are inhibited the mRNA of inflammatory mediators (*iNOS*, IL-6, IFNγ, and TNF-α), increased expression of HSP 70 and 90, and antioxidant proteins, such as HO-1, NQO-1, glutathione reductase, glutathione peroxidase 2, and GST-pi [5]. Also, since interacts with several molecular pathways [4,5,27,28,41,45,56,57,73,74,75,76], it is conceivable that BPC 157 largely interacts with NO-system in various models and species [15] and may induce NO-release of its own, even in the condition where L-arginine is not working [94]. Finally, counteracted oxidative stress [5,49,50,51,55,56,58,72] suggests in the BPC 157 rats the adequate capacity of the oxidant defense system to counteract oxidative stress otherwise regularly presented in the damaged myotendinous junction [78]. Therefore, this means adequate functioning of the antioxidant enzymes in myogenic cells, which not only neutralize excessive ROS, but also affect myogenic regeneration at several stages: influence post-injury inflammatory reaction, enhance viability and proliferation of muscle satellite cells and myoblasts and affect their differentiation [78]. Finally, antioxidant enzymes regulate also processes accompanying muscle regeneration—inducing angiogenesis and reducing fibrosis [78]. Note, BPC 157 has a particular angiogenic effect in wound healing [6,7,18,43,45,46] even in the hyponeural and hypocellular environment (i.e., tendon, fibrocartilage zone) [18,19,20,21,22] that makes the reparative process extremely difficult (since, on the other hand, BPC 157 may heal also corneal ulcer and maintains ocular transparency) [95], stronger angiogenic effect than the standard anti-ulcer agents [46] (note, BPC 157 inhibits the growth of several tumor cell lines and counteracts the effect of vascular endothelial growth factor (VEGF) [74]). Interestingly, in addition to the muscle and tendon healing [18,19,20,21,22], BPC 157 has a particularly beneficial effect against fibrosis also in rat bile duct ligation-induced liver cirrhosis [55]. Finally, the most recent therapy evidence as follow up of the above mentioned recovered endothelium function [47,56], consistently noted in major vessel occlusion studies [52,53,54,57,58,59,60,96], peripherally [52,54,57,58,59,60] or centrally [53], and peripherally and centrally [96], shows that BPC 157 therapy may activate collateral vascular pathways, which are reliant to injury occlusion. Consequently, BPC 157 therapy ameliorated or even counteracted various severe vessel occlusive syndromes [52,53,54,57,58,59,60,96]. Likely, as pointed out [97], this particular beneficial vascular compensatory effect may also essentially contribute to the large wound healing potential including proper healing of both tendon [18,19,20,22,27,28] and muscle [21,35,36,37], and resulting particular myotendinous junction healing.

Concluding, whatever the myotendinous junction lesions current therapy (i.e., biological scaffolds, administration of active compounds, electrospinning, self-reorganized constructs (for review see, i.e., [98])), there is still the unresolved problem of the myotendinous junction and bone-tendon junction as extremely specialized tissues, inadequate repair through mechanically and histologically suboptimal scar tissue, decreased functional properties, and greater risk of recurrent injury [98]. Finally, while the mechanical utilization of contractile force produced by myofilaments requires that they should be efficiently connected to tendon fibers, one of the principal obstacles to the use of cytokines and growth factors in tissue engineering provides delivery vehicles that would localize the factor to the repair site for the relevant period of time and appropriate concentration [98].

Thus, although this study resolving evidence cannot provide precise information about the full mechanisms responsible, the use of stable gastric pentadecapeptide BPC 157 in soft tissue healing (tendon [18,19,20,22] and muscle [21,35,36,37] healing to myotendinous junction healing) should be considered. No cartilage or ossicles formation in other tissues [18,19,20,21,22,35,36,37], particular stability (non-degradation in gastric juice, and thereby intraperitoneal and per-oral application in the present study), and no need for the carrier, easy applicability with no toxicity noted, and large beneficial effects [1,2,3,4,5,6,7,8,9,10,11,12,13,14,15,16,17] can have possible advantages over currently suggested agents and procedures.

## Figures and Tables

**Table 1 biomedicines-09-01547-t001:** Nucleotide sequences.

Gene	Sequence	Size (bp)	
*GAPDH*	TGGCAAGTTCAACGGCACAGT	193	XM_221353
	TTTGGCCTCACCCTTCAGGT		
*eNOS*	CTGGCAAGACCGATTACACGA	206	NM_021838
	TCAGGAGGTCTTGCACATAGG		
*iNOS*	TTGGAGCGAGTTGTGGATTGTTGTTC	126	NM_012611
	GGTGAGGGCTTGCCTGAGTGAGC		
*nNOS*	AACTGGGAGGGGAGAGGATTC	517	NM_052799.1
	GGGTGGGAGGCGAGATTCAT		
*COX-2*	TCAGGAGGTCTTGCACATAGG	157	AF233596
	CTGTATCCCGCCCTGCTGGTG		

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
