# Peer review of "Stable Gastric Pentadecapeptide BPC 157 as a Therapy for the Disable Myotendinous Junctions in Rats"

_biomedicines, 2021, doi:10.3390/biomedicines9111547_

Round 1
Reviewer 1 Report
Dear authors,
Thank you for submitting your article “Satable gastric pentadecapetide BPC 157 as a therapy for the disable myotendinous junctions in rats”.
After reviewing the manuscript, I have to make the following considerations:
- INTRODUCTION Section:
Line 94: Cites 35-27 or 35-37?
- MATERIALS AND METHODS Section:
Line 180: Why skin was colsed with runnion or continuous stitch?. Can this difference in surgical technique influence the results?
Section 2.4. Experimental protocol and assessments: Authors could rewrite this subsection, it is not clear.
Section 2.5: Funtional tests: (Line 217) “To perceive particular aspects of the injured hind limb contracture presentations, rats were awake, anesthetized, or immedially after sacrifice”.
It is not clear whether measurements are made at all 3 times to all rats or measurements are made at different times. If so, please explain if it does not influence the quality of the measurement.
- REFERENCES
Please, ckeck cites 35, 53, 57-60, 62, 69, 71, 72, 90 and 94 are properly worded.
Author Response
Reviewer 1
Open Review
( ) I would not like to sign my review report
(x) I would like to sign my review report
English language and style
( ) Extensive editing of English language and style required
( ) Moderate English changes required
( ) English language and style are fine/minor spell check required
(x) I don't feel qualified to judge about the English language and style
|
Yes |
Can be improved |
Must be improved |
Not applicable |
|
|
Does the introduction provide sufficient background and include all relevant references? |
(x) |
( ) |
( ) |
( ) |
|
Is the research design appropriate? |
(x) |
( ) |
( ) |
( ) |
|
Are the methods adequately described? |
( ) |
( ) |
(x) |
( ) |
|
Are the results clearly presented? |
(x) |
( ) |
( ) |
( ) |
|
Are the conclusions supported by the results? |
(x) |
( ) |
( ) |
( ) |
Comments and Suggestions for Authors
Dear authors,
Thank you for submitting your article “Satable gastric pentadecapetide BPC 157 as a therapy for the disable myotendinous junctions in rats”.
After reviewing the manuscript, I have to make the following considerations:
- INTRODUCTION Section:
Line 94: Cites 35-27 or 35-37?
- MATERIALS AND METHODS Section:
Line 180: Why skin was colsed with runnion or continuous stitch?. Can this difference in surgical technique influence the results?
Section 2.4. Experimental protocol and assessments: Authors could rewrite this subsection, it is not clear.
Section 2.5: Funtional tests: (Line 217) “To perceive particular aspects of the injured hind limb contracture presentations, rats were awake, anesthetized, or immedially after sacrifice”.
It is not clear whether measurements are made at all 3 times to all rats or measurements are made at different times. If so, please explain if it does not influence the quality of the measurement.
- REFERENCES
Please, ckeck cites 35, 53, 57-60, 62, 69, 71, 72, 90 and 94 are properly worded.
Reviewer 2
29 August 2021
Date of this review
06 Sep 2021 11:34:23
Open Review
(x) I would not like to sign my review report
( ) I would like to sign my review report
English language and style
(x) Extensive editing of English language and style required
( ) Moderate English changes required
( ) English language and style are fine/minor spell check required
( ) I don't feel qualified to judge about the English language and style
|
Yes |
Can be improved |
Must be improved |
Not applicable |
|
|
Does the introduction provide sufficient background and include all relevant references? |
( ) |
(x) |
( ) |
( ) |
|
Is the research design appropriate? |
(x) |
( ) |
( ) |
( ) |
|
Are the methods adequately described? |
(x) |
( ) |
( ) |
( ) |
|
Are the results clearly presented? |
( ) |
(x) |
( ) |
( ) |
|
Are the conclusions supported by the results? |
( ) |
(x) |
( ) |
( ) |
Comments and Suggestions for Authors
This is an interesting study to follow the earlier work of the (main) authors however there are numerous points to improve.
Generally, the whole manuscript is longer than it should be, being more concice would be advantegous.
The discussion reviews BPC effects instead of evaluating and connecting the results to the rest of the literature.
The legends and figure panel labellings are negligent:
Fig. 1.: Why two panels (A and B) before therapy? Explain. Columns with dashed line are missing in several panels. The same in Fig. 2.
Fig. 7.: Why it is narower?
Fig. 11.: The naming of BPC treated samples is missing.
Fig 13.: labelling (A,a) is wrong
Fig. 14.: labelling (A,a) is also wrong. Loss of fiber type diversity is apparent on panel “b” but what it is?
Fig. 15.: Column labeling is too small
Gömöri and Masson trichrome stainings show collagen and coll. type III (reticulin) indeed but the implied structure to present knowledge is more general: the whole EC. The authors should interpret it accordingly.
Discussion: l 776-778 elaborating on type III and I coll. ratio contradict to Fig. 11.
To the comments given by the reviewers see our arguments:
Reviewer 1
Dear authors,
Thank you for submitting your article “Satable gastric pentadecapetide BPC 157 as a therapy for the disable myotendinous junctions in rats”.
After reviewing the manuscript, I have to make the following considerations:
- INTRODUCTION Section:
Line 94: Cites 35-27 or 35-37?
Corrected.
- MATERIALS AND METHODS Section:
Line 180: Why skin was colsed with runnion or continuous stitch?. Can this difference in surgical technique influence the results?
In principle, the procedure followed that previously used in our muscle injuries studies (see, i.e., ref. 21). Therefore, it seems to us that this point did not affect the result to any extent.
Section 2.4. Experimental protocol and assessments: Authors could rewrite this subsection, it is not clear.
Acknowledged. It seems to us that the revised version is more clear.
Section 2.5: Funtional tests: (Line 217) “To perceive particular aspects of the injured hind limb contracture presentations, rats were awake, anesthetized, or immedially after sacrifice”.
It is not clear whether measurements are made at all 3 times to all rats or measurements are made at different times. If so, please explain if it does not influence the quality of the measurement.
Acknowledged. The adequate specification is provided. See: To perceive particular aspects of the injured hind limb contracture presentation, measurements were made at all 3 times to all rats while rats were awake, anesthetized, or immediately after sacrifice.
- REFERENCES
Please, ckeck cites 35, 53, 57-60, 62, 69, 71, 72, 90 and 94 are properly worded.
Acknowledged. It seems to us that all of them are ok.

Reviewer 2 Report
This is an interesting study to follow the earlier work of the (main) authors however there are numerous points to improve.
Generally, the whole manuscript is longer than it should be, being more concice would be advantegous.
The discussion reviews BPC effects instead of evaluating and connecting the results to the rest of the literature.
The legends and figure panel labellings are negligent:
Fig. 1.: Why two panels (A and B) before therapy? Explain. Columns with dashed line are missing in several panels. The same in Fig. 2.
Fig. 7.: Why it is narower?
Fig. 11.: The naming of BPC treated samples is missing.
Fig 13.: labelling (A,a) is wrong
Fig. 14.: labelling (A,a) is also wrong. Loss of fiber type diversity is apparent on panel “b” but what it is?
Fig. 15.: Column labeling is too small
Gömöri and Masson trichrome stainings show collagen and coll. type III (reticulin) indeed but the implied structure to present knowledge is more general: the whole EC. The authors should interpret it accordingly.
Discussion: l 776-778 elaborating on type III and I coll. ratio contradict to Fig. 11.
Author Response
Reviewer 2
29 August 2021
Date of this review
06 Sep 2021 11:34:23
Open Review
(x) I would not like to sign my review report
( ) I would like to sign my review report
English language and style
(x) Extensive editing of English language and style required
( ) Moderate English changes required
( ) English language and style are fine/minor spell check required
( ) I don't feel qualified to judge about the English language and style
|
Yes |
Can be improved |
Must be improved |
Not applicable |
|
|
Does the introduction provide sufficient background and include all relevant references? |
( ) |
(x) |
( ) |
( ) |
|
Is the research design appropriate? |
(x) |
( ) |
( ) |
( ) |
|
Are the methods adequately described? |
(x) |
( ) |
( ) |
( ) |
|
Are the results clearly presented? |
( ) |
(x) |
( ) |
( ) |
|
Are the conclusions supported by the results? |
( ) |
(x) |
( ) |
( ) |
Comments and Suggestions for Authors
This is an interesting study to follow the earlier work of the (main) authors however there are numerous points to improve.
Acknowledged.
Generally, the whole manuscript is longer than it should be, being more concice would be advantegous.
Acknowledged. The manuscript is again entirely revised, and few mistakes adequately corrected. We assumed that the „larger“ version will be more instructive for the readers – providing the novelty of the presented myotendinous junction healing. Besides, the most recent evidence, just presented in the Biomedicines, is additionally emphasize in the concluding part, to support the presented findings.
The discussion reviews BPC effects instead of evaluating and connecting the results to the rest of the literature.
Acknowledged. We should emphasize that the current literature of the therapy possible of myotendinous injury was already evaluated in the Introduction,
In the tendon and muscle healing, the lack of the therapy studies upon myotendinous junction injury may be due to the apparent inconsistencies between the suggested various growth factors high physiologic significance [23] and not achieved practical realization of the supposed healing effect [23]. Namely, unlike BPC 157 native peptide therapy [3-17], there is a lacking easy practical applicability for many various growth factors that have been suggested to be necessary for natural healing [23]. The illustrative attempts include both highly sophisticated delivery technics (i.e., a suture carrying nanoparticle/pEGFP-basic fibroblast growth factor (bFGF) and pEGFP-vascular endothelial growth factor A (VEGFA) complexes developed to transfer the growth factor genes into injured tendon tissues to promote healing [24], and various growth factors combinations given together [25,26]), and various carriers (i.e., growth factors-loaded collagen sponge; BMP-12 cDNA-transduced muscle grafts addition [25,26]) to demonstrate experimental usefulness for the tendon or muscle repair, but the direct, local delivery of growth factors has limited use [23].
and that an additional emphasize is made in Discussion (first paragraph).
Regularly, the injured rats definitely remain largely disabled after careful dissecting of quadriceps tendon from the quadriceps muscle, unable to compensate failed crucial knee extensor. Contrarily, the stable gastric pentadecapeptide BPC 157 recovers myotendinous junction. Likely, since with the BPC 157 therapy [19-22,31,35-37,47], the muscle and tendon healing [19-22,31,35-37,47] goes to the myotendinous junction healing, the evidenced osteotendinous junction successfully restored [19,22] may verify a similar restoration of the other junction as well. Together, this brings the myotendinous junction restoration by BPC 157 as the particular healing course, which is obviously a specific and valuable one. This ascertains an essential process not hampered with the initial tendon healing process as similar to fracture healing [83], nor ossicles formation in other tissues as it is with application of the bone morphogenetic proteins (BMPs) [84-86].
Now, additional emphasize was made in the concluding paragraph.
Concluding, whatever the myotendinous junction lesions current therapy (i.e., biological scafolds, administration of active compounds, electrospinning, self-reorganized constructs (for review see, i.e.,[98])), there is still the unresolved problem of the myotendinous junction and bone-tendon junction as extremely specialized tissues, inadequate repair through mechanically and histologically suboptimal scar tissue, decreased functional properties, and greater risk of recurrent injury [98]. Finally, while the mechanical utilization of contractile force produced by myofilaments requires that they should be efficiently connected to tendon fibers, one of the principal obstacles to the use of cytokines and growth factors in tissue engineering provides delivery vehicles that would localize the factor to the repair site for the relevant period of time and appropriate concentration [98].
The legends and figure panel labellings are negligent:
Fig. 1.: Why two panels (A and B) before therapy? Explain. Columns with dashed line are missing in several panels. The same in Fig. 2.
Acknowledged and corrected.
Fig. 7.: Why it is narower?
Acknowledged and corrected.
Fig. 11.: The naming of BPC treated samples is missing.
Acknowledged and corrected.
Fig 13.: labelling (A,a) is wrong
Acknowledged and corrected.
Fig. 14.: labelling (A,a) is also wrong. Loss of fiber type diversity is apparent on panel “b” but what it is?
Acknowledged and corrected.
Fig. 15.: Column labeling is too small
Acknowledged and corrected.
Gömöri and Masson trichrome stainings show collagen and coll. type III (reticulin) indeed but the implied structure to present knowledge is more general: the whole EC. The authors should interpret it accordingly.
Discussion: l 776-778 elaborating on type III and I coll. ratio contradict to Fig. 11.
Acknowledged. To this point see revised Discussion:
Thereby, as a cause-consequence relation appears the counteracted muscle atrophy, no more inflammatory infiltrate in the BPC 157 group, well-oriented recovered tissue of myotendinous junction in BPC 157 treated rats at the 28 days and 42 days. Initially, at the day 7, BPC 157 rats had only small edema and inflammatory infiltrate, connective tissue in more appropriate structure with only a small scar at the 14 days that would not disrupt the orientation as well as continuity of the muscle-tendon junction. As the osteotendinous junction restored (i.e., tendon attaches to bone with an intervening zone of fibrocartilage (direct type) that requests synthesis of strong collagen fibers of type I) [22], goes the other junction, the myotendinous junction, restoration. Type III collagen is also cleared promptly and substituted with collagen type I fibers starting at 14th day of treatment (i.e., improved load to failure correlate with the advanced collagen type I distribution) (Figure 10,e,f,k,l, Figure 11e,f). A synthesis of collagen type I starts before, at 7th day of treatment (Figure 11d), with consequent optimal orientation to the tendon long axis at 14th day of treatment, as consistent evidence based on the specificity of the used methods for collagen staining. In addition to the tendon [18-20,22] and muscle [21,35-37] studies, that finding was also seen in thermally injured animals that received BPC 157 therapy [40,42,44].

Round 2
Reviewer 1 Report
Thank you to authors, they made significant changes in the manuscript following my review report suggestions. The manuscript have improved significantly.
Reviewer 2 Report
Instead of making shorter and more concise the authors add to the length of the text.